# Strategies for the Valorization of Date Fruit and Its Co-Products: A New Ingredient in the Development of Value-Added Foods

**DOI:** 10.3390/foods12071456

**Published:** 2023-03-29

**Authors:** Nuria Muñoz-Tebar, Manuel Viuda-Martos, Jose Manuel Lorenzo, Juana Fernandez-Lopez, Jose Angel Perez-Alvarez

**Affiliations:** 1IPOA Research Group, Centro de Investigación e Innovación Agroalimentaria y Agroambiental, Universidad Miguel Hernández (CIAGRO-UMH), Carretera. Beniel Km 3.2, 033121 Orihuela, Alicante, Spain; nmunoz@umh.es (N.M.-T.); mviuda@umh.es (M.V.-M.); j.fernandez@umh.es (J.F.-L.); 2Centro Tecnológico de la Carne de Galicia, Avd. Galicia 4, 32900 San Cibrao das Viñas, Ourense, Spain; jmlorenzo@ceteca.net

**Keywords:** *Phoenix dactylifera*, date products and co-products, dairy, meat, bakery

## Abstract

Date palm trees (*Phoenix dactylifera* L.) are traditionally cultivated in South-West Asia and North Africa for date fruit consumption, although in recent years, its consumption has increased worldwide, and its cultivation has spread to other areas of America, sub-Saharan Africa, Oceania, and Southern Europe. During date fruit processing, several types of by-products are generated, such as low-quality dates or seeds, which along with date fruit, represent an excellent source of dietary fiber and bioactive compounds such as flavonoids, tannins, carotenoids, tocopherols, and tocotrienols. Therefore, this review provides information on the processing of dates fruit and the value-added by-products generated from them as well as their applications in different types of foods for the development of foods with an enhanced nutritional and functional profile. The incorporation of date fruit and their co-products in food formulations will help to cover the current consumer demands for foods made with ingredients of natural origin and with health properties beyond the merely nutritional.

## 1. Introduction

Dates are the fruits of date palm (*Phoenix dactylifera*), whose cultivation (in arid and semi-arid regions) has important socio-economic, cultural, and dietary values for locals as well as global communities around the world. Nowadays, and due to climate change, more and more regions meet the climatic conditions typical for their cultivation (long and hot summers, low rainfall, and low relative humidity), and so, it is rapidly spreading to other areas in the world [1], with the corresponding expected increase in date fruit production.

Worldwide date production in 2021 extended over a surface of 1,301,979 ha, with total date production of 9,656,377 tons [2], which used to be accompanied by relevant fruit losses during harvesting, processing, storage, and commercialization. It has been estimated that around 30% of date fruits are discarded or used for animal feeding due to their low quality (inadequate texture, unattractive appearance, damaged, blemished, unripe, or undersized) [1,3,4]. In addition, date seeds that represent 10–15% of date fruit fresh weight [5,6] are discarded as waste material after the technological transformation of date fruits or used as animal feed. However, all these date fruit co-products could have many other applications due to their valuable chemical composition.

Date fruits are a rich and inexpensive source of essential nutrients such as carbohydrates (soluble sugars and dietary fiber), minerals, and vitamins [1,7,8], with low levels of lipids and proteins [9]. In addition, date fruits also contain important nutraceutical compounds (phenolic compounds, phenolic acids, cinnamic acid derivatives, flavones, anthocyanidins, isoflavones, and volatile compounds) [6,10] with functional effects, including antioxidant, antimicrobial, anti-inflammatory, antimutagenic, hepato-protective, gastro-protective, anticancer, and immune-stimulatory activities [1,11].

Date seeds are also an interesting source of important nutrients, such as dietary fiber (75–80%), oil (10–12%), proteins (5–6%), and minerals (2–3%) [12,13]. Moreover, their content in some bioactive compounds such as phenolic compounds (21.0–62.0 mg gallic acid equivalents (GAE)/100 g date seeds), total flavonoids, anthocyanins, flavonols, and proanthocyanidins [6,14,15] have increased the interest for their potential functional applications as an antioxidant and antibacterial compounds. Some authors have also reported that date seeds showed inhibitory activities on several enzymes (tyrosinase, α-glucosidase, etc.) [14].

This manuscript reviews the use of date fruits and their co-products as raw materials for the development of date-based foods. It has been separated into two aspects: on the one hand, the products generated after the direct industrialization of the date fruit (or its co-products) to obtain value-added date foods (juice, syrup, flour, paste, etc.) as a way to increase its shelf life and therefore extend the range of its commercialization, and on the other hand, processed foods in which the date fruit (or its co-products) are used as valuable ingredients, analyzing their effect on the different food matrices in which they are incorporated (meat, dairy, starch, etc.).

## 2. Value-Added Date Foods (From Date Fruits and Their Co-Products)

In spite of the interesting nutritional and functional properties of date fruits, they are still seen as a very sweet food, therefore high in calories and to be avoided. However, as has been previously discussed, this is not true since date fruit is not “only sugar" and contains many healthy compounds (bioactive compounds such as polyphenols and dietary fiber) that could be introduced into a healthy diet [16]. This lack of information on the benefits of dates (at least at the consumer level) may be linked to the fact that the main producers of date fruits are developing countries, and their trade with developed countries has been mainly (and almost exclusively) in the form of dried dates (with longer shelf-life but with lower nutritive value and high sugar content) [17]. In order to change this trend, the involvement of both researchers and the food industry with its innovation and development teams is needed. It is easy to check that until now, the date industrialization is relatively scarce (limited to local producers) compared to other fruits or nuts, and it may even be thought that it is underutilized. The industrialization of dates should allow us to extend their availability over time and widen the product offer as value-added date foods. Thus, in addition to their consumption as dried dates (the main marketed product) or fresh dates (local consumption), products such as date juice, date syrup, date sugar, date dietary fiber, etc., should also be available for commercial purposes. In addition, some of the date co-products, mainly their seeds, could be valorized by the industry to obtain food ingredients such as date seed flour, date oil, or date seed extracts rich in antioxidant compounds. Figure 1 shows some of the commercialized value-added date foods.

### 2.1. Date Juice

Date juice is mainly produced from immature date fruits (from low commercial value). For date juice extraction, fresh and firm dates are milled, obtaining a date paste that is preheated and pressed in a hydraulic press. The extracted juice is pasteurized (at 80 °C), cooled, and centrifuged to obtain clear juice. Freshly extracted date juice from India-growth dates containing 19.5% of total soluble solids (TSS), 18.3% total sugars, 13.2 mg/100 g ascorbic acid, 0.38% tannin, and 3.6% pectin was obtained by Kulkarni et al. [18]. These authors reported that using the pectinase enzyme (0.05–0.1 mL/100 g date paste and incubated at 45 °C for 30–150 min) increased the yield and the clarity of date juices. Similar to other fruit juices, date juice has a tendency to lose quality (flavor, vitamins and color) during processing when subjected to heating in open conditions. To avoid this, juices can be concentrated (until 76% of TSS and 67% total sugars) and frozen, achieving the maximum retention of color, flavor, and vitamins, and the concentrated juices could have great demand in the international market. These authors reported that date juice was successfully concentrated using a thin film evaporator in two passes. This concentrate date juice packed in low-density polyethylene pouches, frozen at –40 °C, and stored at –20 °C, was stable for 6 months and could be used for the preparation of ready-to-serve beverages with acceptable sensory quality characteristics. There are not too many references about the production of fresh date juice in comparison with date syrup and sometimes even referred to as juice, but in reality, they are syrups.

### 2.2. Date Syrup

Fruit syrups are concentrated fruit juices widely applied in the food industry and at home as nutritive sweeteners. Second-grade fruits are usually used in order to provide value addition. Date syrup has been produced in local environments and used as a conventional sweetener [19], and even though there is not too much available published information detailing its process, quality characteristics, or shelf-life [19,20], they are mainly limited to traditional processes. For this reason, the need for research in developing date syrup with desirable functional properties that could serve as a substitute for sucrose could be worthwhile [21]. Traditional extraction is carried out with date paste and water at different ratios, being heated at 100 °C several times. The obtained juice is filtered and centrifuged, and then the supernatant is concentrated at 100 °C to 80° Brix, obtaining the syrup. This traditional elaboration process can be improved by the use of pectinolytic and cellulolytic enzymes to increase the extraction yield, reducing sugars and soluble dry matter, as well as reducing turbidity [22,23]. It has been reported that the highest extraction (soluble solids yield (72.37 g of TSS/100 g)), together with the lowest turbidity, was reached using 50 U pectinase/100 g and 5 U cellulase/100 g for 120 min. This syrup also contains a high level of reducing sugar (which could contribute to the reduction of the syrup crystallization phenomenon) and minerals (K: 799.23–1024.78 mg/100 g fresh weight; Ca: 270.55–150.5 mg/100 g fresh weight; Mg: 34.36e78.21 mg/100 g fresh weight; P: 48.3–100.5 mg/100 g fresh weight and Zn: 0.63–1.7 mg/100 g fresh weight). Date syrup, besides its nutritional components, contains various bioactive compounds (phenolic compounds) with antioxidant activity [24]. Recently, Gourchala et al. [25] studied the effect of using whole date fruit (seeds included) for the development of a new syrup formulation. They reported that the incorporation of date pits significantly improved the physicochemical and sensory properties of the newly prepared syrups. In addition, syrups made from entire date fruits showed higher contents of minerals, proteins, polyphenols, and flavonoids than control syrups of pitted dates.

### 2.3. Date Paste

Date paste from fresh non-commercial Medjoul date fruits (date fruits with colors, shapes and sizes that are not demanded by the consumers, dates damaged by insects, etc.) was elaborated on and characterized by Sanchez-Zapata et al. [26]. For that, date fruits were washed and scalded in hot water at 80 °C for 3 min to inactive enzymes. Afterwards, the seeds and the peels were separated, and the flesh part was crushed until a homogeneous paste was obtained. This paste was packed in vacuum pouches and immediately frozen at −30 °C until use. These authors reported that this paste had 35% moisture, 53% total sugars, and 7% TDF as main components. In addition, it showed a total phenolic content of 225 mg GAE/100 g paste. Martín-Sánchez et al. [3] elaborated on an intermediate food ingredient, date paste-type, from Confitera date fruit grown in Spain. For that, they used date fruits at different ripening states (Rutab and Khalal) with and without scalding. The final date paste obtained was shown as a potential source of valuable nutrients, especially reducing sugars (54–82% dw, mainly glucose and fructose), dietary fiber (13.1–15.8% TDF, 10.5–11.1% IDF and 2.3–4.7% SDF), minerals (K: 1118–1285 mg/100 g dw; Mg: 79–80 mg/100 g dw; P: 76–88 mg/100 g dw, and Ca: 24–37 mg/100 g dw), and natural antioxidants (TPC: 4.3 g GAE/100 g TPC). In addition, these authors evaluated some technological properties of date paste in view of its application in the development of date added foods, reporting that date paste could be used as an emulsifying agent (good emulsified activity and stability) for foods requiring emulsion formation and stability during a long shelf-life. Although in both cases the authors have referred to their product as “date paste” from a technological point of view, the elaboration process described is more similar to a date puree. Fruit puree is a form of fruit that has been processed by grinding, squeezing, or mixing one or different types of fruit into a smooth puree with a texture similar to a paste [27].

### 2.4. Date Fiber Concentrates

Date fiber concentrates were obtained by Borchani et al. [28] from date (at the Tamar stage) flesh after extraction with hot water (70 °C, 15 min) and filtration on thin cloth (7 times) to eliminate soluble sugars, proteins, and pectins. Then, the insoluble residue was dried (100 °C, 1 h 30 min) and ground. The same process was applied by Hasnaoui et al. [29] to obtain rich fiber powder from Moroccan date fruits. They reported an extracted yield of total DF in the range of 37–75% (dry matter), depending on the date’s varieties. This dietary fiber concentrate showed moisture of 2.2–5.1% and TDF content of 91–91% dw. This product is not only appreciated for its nutritional value but also for its technological and functional properties. Its water-holding capacity ranged between 4.1–6.2 g water/g dry fiber and the oil-holding capacity between 1.1–1.8 g oil/g dry fiber, which suggests that this concentrate could be used as an ingredient in food and dietetic formulations. In addition, Hasnaoui et al. [29] reported that the fiber concentrate had antiradical capacity (DPPH antiradical efficiency ranging between 1.26 and 7.09 × 10^−3^) and mainly attributed to the polyphenol content, which was considered suitable for antioxidant dietary fiber production. A similar procedure was applied by Fikry et al. [30] to obtain a powder rich in DF. The main processing differences were that, in this case, the number of washings was determined by the TSS content in the washing water (stopped at TSS < 1%) and the drying process conditions (50 °C, 2 m/s air velocity, until a moisture content <9%). These authors reported that the water required for the extraction process of soluble solids was 50–76 L/kg of dry fiber concentrate, with significant variations between date cultivars. A similar water requirement was described by Al-Awaadh [31] to obtain dietary fiber concentrates from Saudi Arabian date fruits. Although date fiber concentrate is a very good source of dietary fiber, its IDF/SDF ratio is very high (regarding the health benefits associated with its consumption). Mrabet et al. [32] investigated and optimized the use of enzymes to enrich their content in SDF. Applying these optimized conditions, the amount of SDF in the dietary fiber concentrate increased from 1.8–6.3 to 5.4 g/100 g, and the ratio of IDF/SDF changed from 19 to 2–3. This SDF, besides an increase in the antiradical activity, contained gluco-, manno-, and xylo-oligosaccharides, so this enzymatic treatment could be a promising process for obtaining tailor-made prebiotic oligosaccharides.

### 2.5. Date Powder

The production of date powder is very interesting and highly beneficial in improving shelf-life, ease of handling, and blendability with foods. Date powder was obtained directly from dates after drying and milling [33]. For that, fresh date fruits were de-seeded and cut into four longitudinal sections and dehydrated (70 °C, 72 h)in a cabinet dryer, and they were then ground and sieved. The quality properties of the resultant powder were: a final moisture content of 2.1–3.7% depending on date variety, water activity in the range of 0.22–0.30, overall solubility ranging from 65 to 82%, hygroscopicity of 10–23 g absorbed water/100 g date powder, and good flowability and low compressibility, both desirable characteristics for powders. These authors also studied the conversion of date powder into cubes/tablets, although due to their low solubility, this option was not totally successful. Other authors reported the production of date powder using spray dryers (150–170 °C and 25–40 mL/min feedstock flow rates) [34]. For that, the use of two carrier agents was investigated (maltodextrin and Arabic gum at 0.4 kg per 1 kg date fruits). Although the physical properties of date powder were significantly affected by the carrier agent, there was no difference in total phenolic compounds between date powders produced with maltodextrin and Arabic gum. Date powder produced with maltodextrin showed smooth, regular-shaped spherical particles but with severe agglomeration. On the contrary, date powder with Arabic gum had relatively smaller particles of irregular spheres with dented surfaces. These date powders could be used as natural sweet powders.

### 2.6. Date Extracts

Different extracts have been obtained from date pulp for their application as antioxidants or antimicrobials due to the presence of phenolic compounds (caffeic acid, ferulic acid, protocatechuic acid, gallic acid, chlorogenic acid, *p*-coumaric acid, resorcinol, catechin, and syringic acid) and flavonoid (quercetin, luteolin, isoquercitrin, apigenin, and rutin) compounds [35,36,37]. These extracts have been obtained directly from date pulp [4,36] or from its defatted powder [38] using different solvents (i.e., water, methanol, ethanol, acetone, or their mix) in continuous extraction apparatus (sometimes ultra-sound assisted) followed by its evaporation under vacuum, and dried to a constant weight using a freeze-drier. The antioxidant and antibacterial activity of these extracts depends on the date cultivar, growth conditions, ripening stage, the solvent used for the extraction, and extraction conditions, among others [37,39]. In all cases, the antioxidant content decreased with increasing fruit ripening, with a corresponding decrease in the antioxidant and antibacterial activities. Most of these authors reported a positive correlation between the antioxidant activities of date fruit extracts and antioxidant contents. Antimicrobial activities against common bacterial food pathogens (*S. aureus, B. cereus, P. aeruginosa, Serratia marcescens,* and *E. coli*) of date extracts have also been correlated with antioxidant contents [4,40]. These date fruit extracts open new promising opportunities for the production of natural antioxidant and antibacterial compounds with potential application in the food industry.

### 2.7. Date Press Cake

Date press cake is a co-product of date fruit juicing that has remained underutilized in the food industry. It is obtained after the industrial extraction of date juice, and it is composed of a mixture of fibrous material from date flesh and smashed pieces of date seeds. Majzoobi et al. [41] reported that date press cake remained after juice extraction from Iran-grown fully ripened Shahani date fruits that contained 13.4% moisture, 4.9% fat, 6.3% protein, 11.7% crude fiber, and 79.1% carbohydrate. Compared to the press cakes from some common fruits, date press cake shows lower fat, protein, and ash content than strawberry and blackberry press cakes but higher fat, ash, and protein content than apple, orange, and pineapple press cakes [42,43,44]. Although its fat content is low, it has an interesting lipid profile with higher amounts of monounsaturated fatty acids than polyunsaturated fatty acids, with oleic acid (C 18:1) being the predominant one. Date press cake contains 17.79 mg GAE/g phenolic content and 1.89 mg quercetin/g flavonoid content. It has been reported that the concentration of phenolic and flavonoid compounds varies depending on the part of the date fruit (flesh, seeds, skin), and so, their content in date press caked is highly related to the juicing process. For example, Al-Farsi et al. [7] reported a higher phenolic content in seedless press cake (2.76 mg/g) and antioxidant activity than in date syrup (1.33 mg/g). The appearance of this date press cake is a powder with a light brown color due to the natural pigments from date flesh and seeds, mainly carotenoids and anthocyanins [45].

### 2.8. Date Seed Oil

The interest in the extraction of oil from date seeds is due not only to its lipid profile (oleic acid 40–50%, linoleic acid 8–19%, lauric acid 18–50%, and palmitic acid 10–15%) but also in its richness in phenolics, tocopherols, and phytosterols [5,46,47]. Due to the hardness of date seeds, oil extraction is hindered, so the application of a pre-treatment is needed (varying the size particle and drying method). Conventional oil-extraction methodology includes Soxhlet extraction and cold extraction, which are straightforward but slow and use large amounts of organic solvents (chemical-consuming processes) [48]. Recently, other environmentally friendly processes have been studied based on hydrothermal pre-treatments (for conditioning the raw material) and ultrasound-assisted techniques or supercritical fluid extraction (for improving the oil yield) [49]. In this sense, oil yield was similar to that obtained by traditional methods (Soxhlet) reported by these authors when some of these environmentally friendly technologies (hydrothermal treatment and sonication) were applied. Another important technological property of the oil is its oxidative stability, and regarding that, Besbes et al. [47] reported that oil date seed subject to Soxhlet extraction showed higher oxidative stability than most vegetable oils and comparable to olive oil. However, this stability was decreased when hydrothermal pre-treatment was applied [49].

### 2.9. Dietary Fiber Concentrate from Date Seed

Date seeds are an excellent source of dietary fiber (50–70%), much better than date pulp [7,50], with a high content of IDF and a low proportion of SDF. To obtain fiber concentrates from date seeds, the same procedures described for date pulp can be applied. These dietary fiber concentrates contain a high percentage of cellulose and hemicellulose (gluco-mannans and galacto-mannans) [50], and their water and oil holding capacity is lower than that reported for dietary fiber from date flesh [50]. In any case, these dietary fiber concentrates can be considered good candidates for the development of fiber-enriched foods. Other authors have developed procedures to fractionate the date seed dietary fiber into water-soluble polysaccharides and hemicellulose and to extract these two fractions separately [12]. For that, it was necessary that date seeds were previously defatted, and the two dietary fiber fractions obtained showed interesting technological properties (water and oil holding capacities and emulsifying properties).

Regarding all these value-added products obtained from date fruits and their co-products, it could be said that almost all of them meet the requirements established to be used in the development of healthy new food products, i.e., they are natural products, allergen-free (suitable for celiacs, or lactose intolerant), with interesting nutritional and functional components that can be concentrated by eco-efficient processes (date paste, date fiber concentrates, etc.) in order to exert a greater physiological effect, or even with technological properties very useful in some food elaboration process, available at a very reasonable price and widely accepted by consumers. All these characteristics have aroused interest in their application as raw materials in the development of new foods (dairy, meat, bakery or pastries, etc.). If we add to this the fact of valorizing the by-products of the agri-food industries, reducing food waste, using local raw materials, and promoting traditional crops, we would be contributing to the UNO/FAO sustainable development goals, especially point 3, “health and well-being, among the promotion of circular economy”. In any case, further controls should be considered to ensure the quality and safety of the by-products used in food formulations.

## 3. Application of Date Fruits Products and Co-Products in Foods

### 3.1. Meat Products

The addition of vegetable or fruit co-products to meat and meat products is an effective strategy accepted by consumers, industry, and the scientific community. In this way, the reformulation of meat products with plant-based extenders (as fruits or their co-products) has given rise to different approaches for obtaining meat products with better nutritional value (reduced calories and increased dietary fiber content) and longer shelf-life (the action of bioactive compounds reduces lipid oxidation and microbial spoilage) [51]. So, the use of plant-based extenders in the development of meat products represents a valuable opportunity not only for the addition of healthier ingredients but also reducing production costs [52]. Vegetable and fruit co-products could be considered non-meat ingredients added to meat products as extenders which have a high content of proteins and dietary fiber that can also improve several physico-chemical, techno-functional, and sensorial properties of products, including water-holding and emulsion capacities, textural parameters such as hardness, chewiness, and springiness, flavor, palatability, and overall appearance [53].

The addition of date fruit co-products into meat products is widely established. Several researchers have reported the effect of the reformulation of meat products using date fruit co-products in fresh, cooked-cured, and dry-cured meat products. In this sense, date seeds have been used as an ingredient in several meat products enhancing the nutritional value by the addition of insoluble dietary fibers, reducing calories, extending meat product shelf-life (due to the bioactive compounds), and improving the texture [54,55,56,57]. Thus, in fresh meat products such as burgers, Sayas-Barberá et al. [55] analyzed the addition of date seed powder, at different concentrations (1.5, 3.0, and 6.0%), into beef burgers. These authors reported that the incorporation of date seed powder improved the cooking properties as well as the sensorial acceptability of the burgers. In addition, the burgers added with date seed powder had lower lipid oxidation values and microbial counts than the control burger after 10 days of storage. Abdel-Maksoud et al. [57] carried out a work to assess the quality and antioxidant properties of beef meatballs added with date seeds powder (0, 4, 8, 12, and 16%). These authors found that date seeds powder addition in beef meatballs improved the nutritional profile (higher contents of crude fiber, ash, and total carbohydrate with reduced fat content), phenolic content, and antioxidant activity. Additionally, technological properties, including water-holding capacity, cooking yield, and cooking loss, were enhanced without significant changes in the organoleptic properties of the beef meatballs. Alqahtan et al. [58] conducted a study to analyze, in beef burgers, the effects of replacing the breadcrumbs (25, 50, 75, and 100%) with date seed powder obtained from dates in the first stage of ripening. They reported that this substitution reduced moisture, protein, and lipid content in burgers with respect to control samples (depending on concentration). Similar effects were observed for the textural parameters such as hardness, gumminess, and chewiness, whose values were lower in burgers with date seed powder. On the other hand, the substitution of breadcrumbs with date seed powder at 25% was the most suitable concentration since lipid oxidation was prevented, microbial counts were reduced, and cooking properties were improved. In addition, the beef burgers with 25% substitution reached the highest acceptability in the sensorial assay.

Date seeds may be used to marinate fresh meat cuts. Thus, in chilled chicken, Abdelrahman et al. [56] informed us that the treatment of samples with 2% date seeds extract had lower total volatile nitrogen and TBARs values as compared with control ones due to their content of bioactive compounds. Additionally, this treatment with date seed extracts significantly reduced the aerobic bacterial counts and extended the shelf-life of stored chicken meat, while *Salmonella* spp. and *Campylobacter* spp. were not detected in treated samples. Similarly, Nor et al. [59] evaluated the tendering effect of date seed powder towards beef and lamb meat. These authors found that textural properties such as hardness, springiness, chewiness, and gumminess decreased between the different amounts of seed powder used in marinated beef and lamb with respect to control. In addition, cooking loss and water holding capacity was decreased when the date seed powder content was increased. Date seed powder may also be used to replace fat in meat products. In this sense, Essa and Elsebaie [54] elaborated that beef burgers contained date seeds powder as a fat replacer (substitution ratio of 25, 50, and 75%). These authors informed us that in reformulated samples, the cooking yield, fat retention, and moisture increased with respect to controls. Additionally, these burgers were more resistant to lipid oxidation and showed better nutritional value than the control without showing any negative effects on the sensory properties. In more recent and interesting research, Essa and Elsebaie [60] elaborated on a gel composed of date seed powder and gelatin to be used for making low-fat burgers. The addition of this gel produced a reduction in the hardness and chewiness values of low-fat burgers whilst the springiness was increased. In addition, the use of this gel as a fat replacer raised the moisture, ash, protein, and some minerals, such as the sodium and calcium content of the burger, and improved the technological properties, including coking yield, fat retention, and water retention. Previously, Bouaziz et al. [61] formulated low-fat turkey burgers added with Deglet Nour date seeds flour at 3, 5, and 10%. They informed that burgers elaborated with date seed flour, up to 5%, had similar textural properties and improved sensory parameters (texture, flavor, and overall acceptance) than control samples. Additional studies about the use of date fruit and its co-products on meat products are shown in Table 1.

### 3.2. Dairy Products

Consumers have changed their eating habits in recent years, increasing the demand for improved healthy foods such as functional foods or those whose nutritional profile has been modified (e.g., fat-low foods or enriched with fiber). In this sense, the dairy industry is probably one of the most traditional, providing evidence of the need for innovation in order to meet the current demands and stand out in an increasingly competitive market. Milk and dairy products are among the most consumed foods due to their protein, vitamin, and mineral content, making them one of the main food groups for the incorporation of new ingredients for functional purposes.

As previously mentioned, dates are a fruit with an excellent nutritional profile and a source of several functional compounds related to numerous benefits for human health and disease prevention. Such is the case of dietary fiber that promotes the colonization of beneficial bacteria, such as *Bifidobacterium* spp. and *Lactobacillus* spp., contributing to maintaining optimal intestinal function [68]. In addition, fiber content is highly valued by food industries in the development and innovation of new foods because it has therapeutic benefits such as anti-diabetic properties, limited cholesterol absorption, or generation of short-chain fatty acids [9]. Likewise, date fruit and its by-products are a good source of dietary fiber, providing antioxidant and antimicrobial activity due to its lignin and tannin composition [69]. Therefore, the incorporation of date products or by-products in dairy products can be a good and suitable strategy for the development of new functional foods with improved health benefits and added value, as well as helping the revalorization of date industrial by-products promoting circular economy in the food industry. As can be seen in Table 2, dates have been incorporated into dairy products obtaining an improved quality and nutritional profile as well as a better antioxidant capacity and good sensory properties.

Djaoud et al. [83] formulated a new dairy dessert incorporating date by-products (syrup and powder) as a sweetener to substitute sugar addition. The results of the study showed that the substitution of 16% of sugar in dairy desserts with syrup and date powder as sweetening ingredients is a feasible option since their incorporation does not affect the microbiological quality of the dairy product. The incorporation of these date by-products resulted in an improvement of the antioxidant capacity, polyphenol content, and composition (lipids, proteins, and dry matter) of the formulated dairy desserts, as well as the absence of protein cross-linking. Likewise, the structure observed in the CONFOCAL revealed that the formulated desserts displayed a homogeneous structure. This study demonstrated that the substitution of sugar for syrup and date powder could be a useful alternative for the development of new functional products with good consumer acceptance. Date syrup and date powder have also been incorporated into other dairy products, such as cheese or fermented milk. For instance, a spreadable processed cheese was supplemented with roasted date seed powder at different concentrations (1, 5, and 10%). The addition of this date by-product resulted in a reduction of protein, soluble nitrogen, and ash, while total solids, fat, and fiber increased. Likewise, the authors reported a proportional increase in mineral content (Na, K, Fe, and Zn) compared to the control samples and a higher content of total polyphenols and flavonoids. This work also showed that the meltability and oil separation index decreased over storage time when date seed powder was added and that the samples obtained good sensory acceptability [84]. Alqattan et al. [85] evaluated the use of this date by-product as a promising and novel fat replacer and fiber source in processed cheese block types. Four concentrations of date seed powder (5, 10, 15, and 20%) of cheese fat replacements were used to evaluate their effect on chemical composition, microstructure, rheology, and sensory properties. The results showed that the incorporation of this extract improved the fiber content and texture of the reformulated cheeses in addition to stabilizing the hardness, adhesiveness, and springiness. The microstructure of the cheeses containing the date pit powder presented fewer fat globules and a homogeneous and evenly distributed protein network compared to the control. Samples with 5% fat replacement received the closest scores to the control in the sensory analysis, and this work proved that the use of date pit powder could be effectively used as a fat replacer in the development of healthier cheeses with a similar texture and unaffected microstructure.

In relation to the incorporation of date syrup in fermented milk, a fermented milk beverage (Laban) flavored with date syrup (dibs) was developed at different concentrations (2.5, 5, 7.5, 10, 12.5, and 15% date syrup/total weight of Laban) selecting the 12.5% concentration (74 ^◦^Bx) as the most suitable after a sensory preference analysis with panelists. The study evaluated the nutritional profile, microbiology, and sensory quality of this fermented milk for 7 days in cold storage at 4 °C. Flavored Laban incorporated with date syrup showed an increase in pH, ash, protein, total solids, sugars, and magnesium whilst acidity, fat, casein, lactose, calcium, total microbial count, and total yeast and molds count decreased at the end of the study. Sensory evaluation revealed that the flavored Laban drink was more acceptable than the control samples after 7 days of cold storage [86]. In a more recent study, Shahein et al. [87] produced probiotic fermented camel milk with date syrup as a prebiotic and flavoring agent to evaluate its physicochemical, phytochemical, microbiological, and sensory properties after 1 day and 15 days of storage at 5 °C. Date syrup was added at 6 and 8%, and fermented non-flavored camel milk served as a control. The incorporation of date syrup significantly increased the total solids, ash, minerals (Na, K, and Fe), total phenolic content, viscosity, and antioxidant values of the fermented camel milk. Total bacteria and *Bifidobacteria* counts increased with the concentration of date syrup, and sensory scores for flavor, consistency, appearance, and total scores were improved compared to the control samples.

As previously described, date juice obtained from unripe fruits can be a good option as a novel ingredient in the development of dairy products due to its composition. In this sense, a study was recently carried out on the effect of its incorporation (2, 5, and 10%) in the physicochemical and microbiological characteristics of a bio-yoghurt (*Streptococcus thermophilus*, *Lactobacillus acidophilus*, and *Bifidobacterium longum* as a probiotic starter) [88]. Yogurts formulated with date juice presented an increased acidification rate and a lower pH value, as well as improved syneresis. During the 21 days of the study period, the viability of probiotics decreased, but a higher number of probiotic bacteria was observed in the samples containing the highest concentration of date juice compared to the rest of the samples.

Among the diverse by-products generated from date plants are spikelets, which contain small unripe date fruits in their earliest stage of development. During fruit thinning carried out to improve the quality and reduce the tree load, approximately 4–5 racemes containing roughly 25–100 spikelets are removed [89]. The study conducted by Almusallam et al. [90] revealed that a date palm spikelet extract obtained from Reziz and Khalas cultivars presented high amounts of total phenolic and flavonoid with DPPH and ABTS radical scavenging activity exceeding 94%. Therefore, in a later work, these authors incorporated this extract at 0.5 and 1% in cow’s milk yogurts to evaluate its physicochemical and microbiological properties during cold storage. Spikelet extract caused an increase in the total polyphenol and flavonoid content of the yogurts, which resulted in a higher antioxidant capacity (DPPH from 10.87 to 63.12–75.33% and ABTS from 18.22 to 37.73–42.81%) and reduced lipid oxidation with lower TBARs values (from 2.50 to 1.13–1.25 mg MDA/kg) at 21 days of storage. Yogurts formulated with this extract showed higher scores in overall acceptability than the control as well as a stable gel matrix and adequate LAB growth [89].

Last but not least, date paste obtained from date flesh is mainly composed of sugars (fructose, glucose, and sucrose), dietary fiber, and polyphenols, so its incorporation in dairy products as a sweetener can be a good alternative to the use of refined sugars. In this sense, fermented camel milk was incorporated with Sukkari date paste in different concentrations (5, 7.5, 10, 12.5, and 15%) to evaluate the effect on physicochemical, rheological, and organoleptic properties as well as on the viability of probiotic strains during 15 days at cold storage [91]. The addition of date paste resulted in an increase in total solids, ash, dietary fiber (from 0 to 0.64 g/100 g), and carbohydrates (3.7 vs. 13.67 g/100 g) compared to plain fermented camel milk along with an increase in mineral content (K, P, Mg, Zn, Fe, and Cu) and antioxidant activity. Probiotic strains (*Streptococcus thermophiles*, *Lactobacillus acidophilus*, and *Bifidobacterium bifidum*) viability improved when low and medium date paste concentrations were incorporated. Fermented milk samples with 10 and 12.5% of Sukkari date paste had the best balance flavor score at the beginning and end of storage, respectively. The authors concluded that the incorporation of date paste from this variety at percentages between 7.5 and 12.5% enhanced the nutritional quality of the fermented camel milk without impairing the viability of the probiotics and their technological and sensory characteristics. Similarly, Tawfek et al. [92] studied the effect of date paste (10, 20, and 30%) incorporation in the composition and texture of fermented camel and goat milk. In this study, the authors observed that pH, protein, and fat decreased while total solids, carbohydrates, ash, vitamins, and minerals increased. In addition, the viscosity and antioxidant capacity of the reformulated fermented milk were also increased. These authors reported higher microbial counts in both fermented goats’ milk and camels’ milk incorporated with date paste. Both products displayed good sensory acceptance (freshly and through storage), proving that this date product can be a suitable and good sweetener as an alternative to refined sugars.

### 3.3. Bakery Products

Bakery products are among the most widely consumed foods, in big amounts on a daily base, having an important role in human nutrition. Fortification of bakery products with functional ingredients in order to improve their health properties and nutritional profile is a commonly used approach to fulfill current consumer demands. The principal ingredient of bakery products is wheat flour, which is responsible for the structure and the bulk of the product, followed by water, yeast, salt, and other ingredients. However, the increasing demand for foods with additional health benefits has promoted the inclusion of bioactive compounds or dietary fibers as new ingredients, representing one of the most researched and used trends in recent years.

People’s current busy lifestyles often lead them to resort to less healthy ready-to-eat foods that may increase the risk of developing diseases such as diabetes, obesity, cardiovascular diseases, and high blood pressure. Therefore, fortifying bakery products with bioactive ingredients and dietary fibers can help minimize these health problems related to poor dietary habits [93].

In the case of fiber, it is well known that the population does not consume sufficient amounts of fiber on a daily basis, so the enrichment of bakery products with dietary fiber could help to increase the recommended fiber intake. Likewise, the incorporation of fiber reduces blood glucose levels due to digestible carbohydrates, making it possible for patients with diabetes to consume bread or bakery products [93]. In relation to the bioactive compounds obtained from dates, they can be used as a more natural alternative for synthetic antioxidants such as butylated hydroxyanisole (BHA) and butylated hydroxytoluene (BHT) so that the toxicity levels and process cost can be lowered [94].

Nowadays, the valorization of by-products from the food industry by reincorporating them into the food chain is one of the main approaches used to improve the sustainability of food production through the reduction of the amount of waste generated, thus promoting the circular economy. In this sense, the processing of dates for the manufacture of value-added food products, including syrup, paste, jam, jelly, and vinegar, produces various co-products, such as seeds that are rich in flavonoids, phenolic acids, tocopherols, and phytosterols [5,13,14] that can be incorporated in bread and bakery products for functional purposes. Moreover, date by-products have nutrients for microbial growth, being a good substrate for the production of fermentation products, including organic acids and polysaccharides [93,95].

In the past decades, there has been an evolving trend to use dates and their co-products in bread and bakery products to improve their nutritional profile and functional quality. Some of these studies are shown in Table 3, highlighting the co-product used, the food to which it has been added, and the main effects caused.

The potential of date fruit pulp to substitute sugar in the elaboration of cookies was investigated by Akhobakoh et al. [104]. These authors studied the effect of substituting sugar for date pulp and eggs for bean milk on two cookie formulations, observing that the highest protein content was obtained in the cookies containing date pulp (1.78%). Furthermore, the fortification of cookies with date pulp increased the iron and zinc content (39.95 and 13.65 mg/100 g, respectively) compared to the rest of the formulated cookies. Sensory evaluation showed that the reformulated date cookies were rated overall best in saltiness, acidity, crispiness, creaminess, and bitterness, demonstrating that date pulp powder can be a suitable replacer of sugar. Similarly, Turki et al. [105] developed biscuits formulated with date powder to improve their functional properties. In this work, the powder of two varieties of date (*Phoenix canariensis* Hort. Ex Chabaud, red and yellow) was incorporated (9 and 7%). The incorporation of date powder in the cookies resulted in an increase in hardness, polyphenols, fiber, and antioxidant capacity. The biscuits were formulated with a 25/75 mix of red/yellow date powder and were the most overall liked. Najjar et al. [106] studied the substitution of wheat flour for ground date seed (2.5, 5.0, and 7.5%) in a cookie’s formulation to enhance its antioxidant capacity. The results showed that cookies with date seed powder contained significant amounts of TPC and flavonoids, leading to an increase in the antioxidant capacity in comparison with the control cookies without date seed powder.

Regarding the incorporation of date in bread, Bouaziz et al. [11] investigated the effect of date seed (concentrations 0.5% and 0.75%) as a dietary fiber source to improve wheat bread’s quality. Date seed showed good functional properties in wheat flour with minimal bread-making quality. Finally, the incorporation of date syrup in bakery products as a sugar replacer was investigated by Lajnef et al. [107] to formulate a sponge cake. These authors evaluated the effect of sucrose substitution at three levels (50, 75, and 100%), observing that the sponge cakes had a similar texture to the control, with adhesiveness and chewiness as the most affected parameters. The antioxidant capacity and TPC were increased in the formulations containing date syrup instead of sucrose, and the color was darker, reddish, and less yellow as the date syrup content increased. This study demonstrates that sucrose can be completely substituted by date syrup in the formulation of this type of bakery product, resulting in a lower caloric cake with a healthy profile due to the minerals and polyphenols of dates.

## 4. Conclusions

Based on current health concerns and a preference for naturally flavored foods, dates represent a promising alternative to replace chemical additives, giving products good nutritional properties and good acceptability by consumers. The information gathered in the present review on the different applications of dates and their co-products in different foods demonstrates that it is a very versatile fruit whose revaluation can help the palm industry by producing in a more sustainable manner, promoting the circular economy in the food chain, and increasing their profits.

## Figures and Tables

**Figure 1 foods-12-01456-f001:**
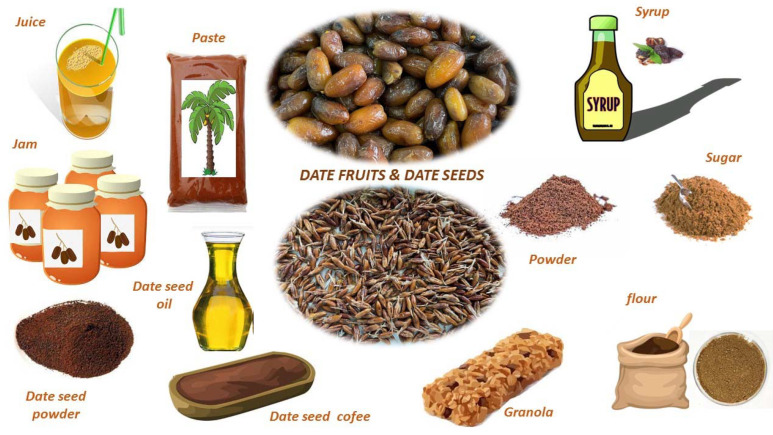
Variety of food products based on date fruit and co-products.

**Table 1 foods-12-01456-t001:** Summary of different studies about the incorporation of date palm and its co-products in meat products.

Date	Concentration	Food	Effect	Reference
Date paste	2.5 and 7.5%	Spreadable liver pâtés	<hardness and gumminess; >moisture and fiber content; <protein, fat and nitrite content; >emulsion stability; softer pâtés; acceptable sensory quality.	[62]
Date pit	0.50, 0.75, and 1.00%	Ground beef	<TBAR values at the end of the storage period (from 2 to 0.69 mg MDA/kg); >values of taste, odor, color, and overall acceptance.	[63]
Date paste	5.0,10.0, and 15.0%	Pork liver pâté	Pâté with 5% date paste > color stability during storage; pâté with 10% date paste < lipid oxidation; similar texture; >overall acceptance	[64]
Date paste	5.0, 10.0, and 15.0%	Bologna sausage	Better nutritional profile (<fat and > fiber content); >red color; <hard, chewy and cohesive; satisfactory sensory quality	[25]
Date paste	2.5 and 7.5%	Pork liver pâté	>phenolic compounds content during storage (the highest at 7.5%); <mesophilic aerobic bacteria (from 3.81 to 2.48 log CFU/g) at 21 days of storage.	[65]
Date paste (intermediate food product)	5%	Paprika dry-cured sausage	Similar ripening process and sensory quality	[66]
Date powder	1.25, 2.5, 3.75, and 5.0%	Sausage	>nutritional properties (<fat content); <residual nitrite level; <lipid oxidation; <Lightness and redness; similar yellowness; >overall acceptability (5% the highest).	[67]

**Table 2 foods-12-01456-t002:** Summary of different studies about the application of date fruit and its co-products in dairy products.

Date	Concentration	Food	Effect	Reference
Date syrup and powder	3 formulations with different powder/syrup ratios: – enriched in powder with powder/syrup = 2 – equal weight mixture of powder and syrup – enriched in syrup with powder/syrup = 0.5	Dairy dessert	Improved apparent viscosity and spontaneous exudation; >antioxidant activity; samples with Kentichi and Deglet Nour varieties were the most appreciated by consumers for their taste, flavor, and texture.	[70]
Date syrup	2.0, 4.0, 6.0, 8.0, and 10.0%	Yogurt	>total phenolics content and pleasant flavor (syrup 10% syrup the highest); >antioxidant capacity and folate concentration at 10% syrup added.	[71]
Date pulp and date liquid sugar	Sugar substitution (50 or 100%)	Ice cream	>viscosity, total phenols and anthocyanin content; >antioxidant capacity (DPPH scavenging from 18.14 to 41.45%, and reducing power 0.23 vs. 1.63 BHT equivalent/100 g sample); flavor enhanced.	[72]
Date syrup	4.0, 7.0, and 10.0%	Prebiotic chocolate milk	>overall acceptability at higher concentrations	[73]
Date paste and date syrup	Substitution of sugar at different levels (25, 50, 75, and 100%)	Kesari (Indian dairy dessert)	similar texture profile	[74]
Date syrup	Sugar replacement by 20, 40, 60, and 100%	Ice cream	>viscosity; 60% replacement is the best acceptance after the control sample.	[75]
Date flour	1.0, 2.0, 3.0, and 4.0%	Ice cream	>viscosity; 1 and 2% the best sensorial quality; <fat content(from 4.63 to 3.86–4.17%); >K, S, Mg, Fe, Zn and Ni content	[76]
Date syrup	1.0, 2.0, and 3.0%	Fermented milk	2% date syrup > flavor, appearance and texture scores); >total solids and total amino acids content.	[77]
Date blanching water	15 g/100 mL	Yogurt	Confitera blanching water produces soft gels of weak structure; Medjoul > firmness and better sensory scores; >lactic acid in Confitera; >lactose content.	[78]
Date syrup and date powder	3, 5, and 10 g	Mold-ripened soft cheese	>total solids and protein content; similar growth of *S. thermophilus*, *G. candidum* and *P. camemberti*	[79]
Date extract	4.0, 8.0, and 12.0%	Fermented milk	>Total solids (from 9.98 to 16.88 g/100 g); >ferric reducing power (from 4.01 to 22.24 mg ascorbic acid equivalent/100 g); similar sensory acceptability	[80]
Date paste	5, 10, 15, and 20%	Yogurt	>Total solids, pH, total phenolic content and DPPH radical scavenging activity; <acidity and syneresis; >sensory evaluation score (15% > 10%)	[81]
Date palm pomace syrup	2, 4, and 6%	Yogurt	<syneresis; improved texture; similar sensory characteristics; >viscosity; >mineral (K, Ca, P, Mg, Fe, and Zn) content	[82]

**Table 3 foods-12-01456-t003:** Summary of different studies about the application of date fruit and its co-products in bakery products.

Date	Concentration	Food	Effect	Reference
Date fruit pulp	100:0, 75:25, 50:50, 25:75, and 0:100 (sugar:date)	Bread	>nutritional quality (>protein, crude fiber and fat content)	[96]
Date seed powder	5, 10, 15, and 20%	Pita bread	>Fiber content (up to 8.94 % in 20% date seed addition); >antioxidant capacity; <acrylamide content (>50% reduction)	[97]
Date seed flour or date seed flour hydrolysate	2.0 and 5.0% or 2.5%	Muffin	Date seed hydrolysate: improves texture; >moisture content and acceptation; >angiotensin I converting enzyme inhibition with an IC50 value of 16.7 mg. Date seed flour: >dark brown; <sensory acceptance; >total dietary fiber content.	[98]
Date fruit fiber	2.5 and 5.0%	Muffin	>dietary fiber (from 1.88% to 2.24–2.55%); >antioxidant capacity; softer muffin; good scores in the sensory evaluation	[99]
Defatted date seed	1.0 and 3.0%	Bread	>dietary fiber content; similar sensory quality	[100]
Date syrup	25, 50, 75, and 100% (instead of sugar)	Flakes	Similar nutritional profile and consumer acceptance	[101]
Date powder	Wheat flour, chickpea flour and date powder in different ratios: 100:0:0, 80:10:10, 60:20:20, 40:30:30, and 20:40:40	Biscuits	>fiber and protein content; substitution up to 30% chickpea flour and 30% date powder the most acceptable biscuits (color); <sugar levels (nearly 60%)	[102]
Date powder	5,10,15,20,25,30,35 and 40%	Gluten-free biscuits	<diameter, thickness, weight, spread ratio, density, and hardness; >darkness; >ash, fiber, and carbohydrate content; up to 20% similar sensory quality.	[103]

## Data Availability

The date presented in this paper are available upon request from the corresponding author.

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
