# Peer review of "Strategies for the Valorization of Date Fruit and Its Co-Products: A New Ingredient in the Development of Value-Added Foods"

_foods, 2023, doi:10.3390/foods12071456_

Round 1

Reviewer 1 Report

Dear Author,

I wanted to take a moment to express my gratitude for your recent review. Your writing is clearly the result of a lot of careful thought and attention to detail, and I appreciate how you managed to capture all of the ideas presented in the subject.

Your review was not only informative but also engaging, making it easy to follow along and understand the key points being made. Your insights and analysis were also especially valuable, and I feel that I learned a lot from your perspective.

my comments were sticky on attached pdf file, which refer to some statics around the production of dates, as well suggested references maybe useful for your review.

Once again, thank you for your thoughtful work.

  1. The main question addressed by the research is the potential of date palm tree by-products as a source of dietary fiber and bioactive compounds, and their applications in the development of functional foods.
  2. The topic is relevant in the field as it focuses on a sustainable and underutilized source of bioactive compounds and dietary fiber. The review highlights the potential of date palm tree by-products in developing functional foods with enhanced nutritional and health benefits. The review also addresses the gap in the literature on the applications of date palm tree by-products in the food industry.
  3. The review adds value to the subject area by providing a comprehensive overview of the processing of date fruit and the by-products generated from them, along with their nutritional and functional properties. The review also discusses the potential applications of these by-products in the development of functional foods, which is an emerging trend in the food industry.
  4. The authors could consider providing more information on the extraction and characterization methods of the bioactive compounds in the date palm tree by-products.
  5. Further controls should be considered to ensure the quality and safety of the by-products used in food formulations.
  6. The conclusions are consistent with the evidence and arguments presented in the review. The authors suggest that the incorporation of date palm tree by-products in food formulations can lead to the development of functional foods with enhanced nutritional and health benefits.
  7. The references cited in the review appear to be appropriate and relevant to the topic discussed.
  8. The tables and figures included in the review are clear and informative, providing useful information on the nutritional and functional properties of date palm tree by-products. However, it would be helpful if the authors could provide more detailed descriptions of the figures and tables in the main text.

Regards

Reviewer 2 Report

Dear Authors,

The manuscript is very relevant today. addresses the issue of nutritional value and the use of waste materials, which is part of the circular economy.

The manuscript in a well-edited and very essential way describes the possibilities of using dates, waste management and nutritional values that we can get by using them.

I believe that the work has been prepared very reliably, it contains minor editorial and linguistic errors.

75% of the literature is from the last 10 years.

Reviewer 3 Report

The review entitled “Strategies for the valorization of date fruit and its coproducts: A new ingredient in the development of value-added foods” reports interesting and detailed information about the valorization of date fruit and its co-products. The review is written well, and the information is ordered. The references used were adequate for the information provided. 

Please refer to the attachment for suggestions to the author.
